

# Re-Analysis of one of the deadliest Tornadoes in European History and its implications

Alois M. Holzer[1], Thomas M. E. Schreiner[1, 2] and Tomáš Púčik[1]

[1]ESSL, European Severe Storms Laboratory, Wiener Neustadt, 2700, Austria
[2]Skywarn Austria, Steyr, 4400, Austria

*Correspondence to*: Alois M. Holzer (alois.holzer@essl.org)

**Abstract.** Extremely rare events with high potential impact, like tornadoes, are of strong interest for climatology and risk assessment - especially though not only in the context of climate change - but difficult to tackle statistically. In order to

widen the data basis especially for the scarcest high-end tornadoes, it is vital to study historical events. It could be speculated that a windstorm catastrophe that happened about 100 years ago is difficult to re-analyze, as the one that happened in Wiener Neustadt, Lower Austria, on 10 July 1916. The present paper demonstrates feasibility in the given case and proposes a working method for similar requirements.

After presenting the methodology, a chronological description of the environmental atmospheric conditions and of the

tornado itself is given, followed by results and new findings compared to a historical scientific study that was published soon after the event. Our analyses found complex thunderstorm activity in the study area and revealed an untypical tornado genesis – with only single parallels in US severe storms literature of the 20[th] century. For the urban area of Wiener Neustadt rich data sources allowed us to draw a detailed picture of damage tracks and hence wind strength in the tornado. Also the originally stated number of fatalities needs to be corrected upwards.

In the outlook we postulate the requirement for an International Fujita Scale to rate tornadoes worldwide in a consistent way. Comparable damage indicators are required for this to be useable. We also stress the side-benefits of the given study for the field of civil protection, namely to serve as an exercise scenario and as a basis for public awareness rising.

## 1 Introduction

Before this study, on a regional level, and in scientific literature also on a national level, it was well known that a

catastrophic windstorm event took place in Wiener Neustadt, Lower Austria, on 10 July 1916, causing a high number of fatalities and injuries. The event was called "Windhose" or "Wirbelsturm" or "Orkan" in the contemporary German text sources. While in 1916 the term "Wirbelsturm" was well understood to name the phenomenon we call "tornado" today in the language of lay people, in later decades the term was confused with tropical cyclones. In contrast, the important book "Wind- und Wasserhosen in Europa" (Wegener, 1917) did not see a direct synonym for the word "tornado". In German

language scientific literature of the early 20[th] century the term "Großtrombe" was most often used to describe a tornado.



Such a conclusion was not clearly drawn in the original literature of the given event, leaving it unclear whether the event would be classified as "tornado" according to nowadays definition.

In order to allow for a sound classification, a detailed re-analysis is needed, presupposing that there is enough original
information available to perform such. In the given case both written reports and a few damage photographs of the severe windstorm event were known already before our study was performed. In 1917 a detailed scientific paper (Dörr, 1917) documented the event and its meteorological interpretation. This paper included information retrieved from eye-witnesses and a city map of the buildings damaged or destroyed. Contemporary knowledge didn't allow for estimations of the maximum strength of a windstorm or tornado, but after 100 years we are now in the position to attempt this goal, given a
much better general knowledge about severe convective storms and tornadoes.

The assessment of extreme events is important for sound statistics of those rare phenomena. This is especially true for the frequency of violent tornadoes and their climatology. Tornado events around Wiener Neustadt per se are not unknown. In fact it seems that around this town a local maximum is present in the available data (Groenemeijer and Kühne, 2014).
Tornadoes in the region of interest around Wiener Neustadt - defined as the area between the geographical coordinates 47.65N/15.85E and 48.20N/16.65E - were collected not only by Alfred Wegener, but also by Johannes Letzmann (Letzmann, 1939), Anton Pühringer (Pühringer, 1973), and finally Otto Svabik and Alois M. Holzer (Svabik and Holzer, 2005), as it was similarly the case in other parts of Europe (Antonescu et al., 2016). A recent study on an outstanding historical tornado in the USA was published on the so-called TriState tornado (Johns et al., 2013).
The European Severe Weather Database, ESWD (Dotzek et al., 2009), was established in 2006. The ESWD collects and rates both historic and recent events and provides for the region of Wiener Neustadt now a continuous dataset of events from the beginning of the 19[th] century to present. In total 23 confirmed tornado events are found in the ESWD (Fig. 1) within the defined area.



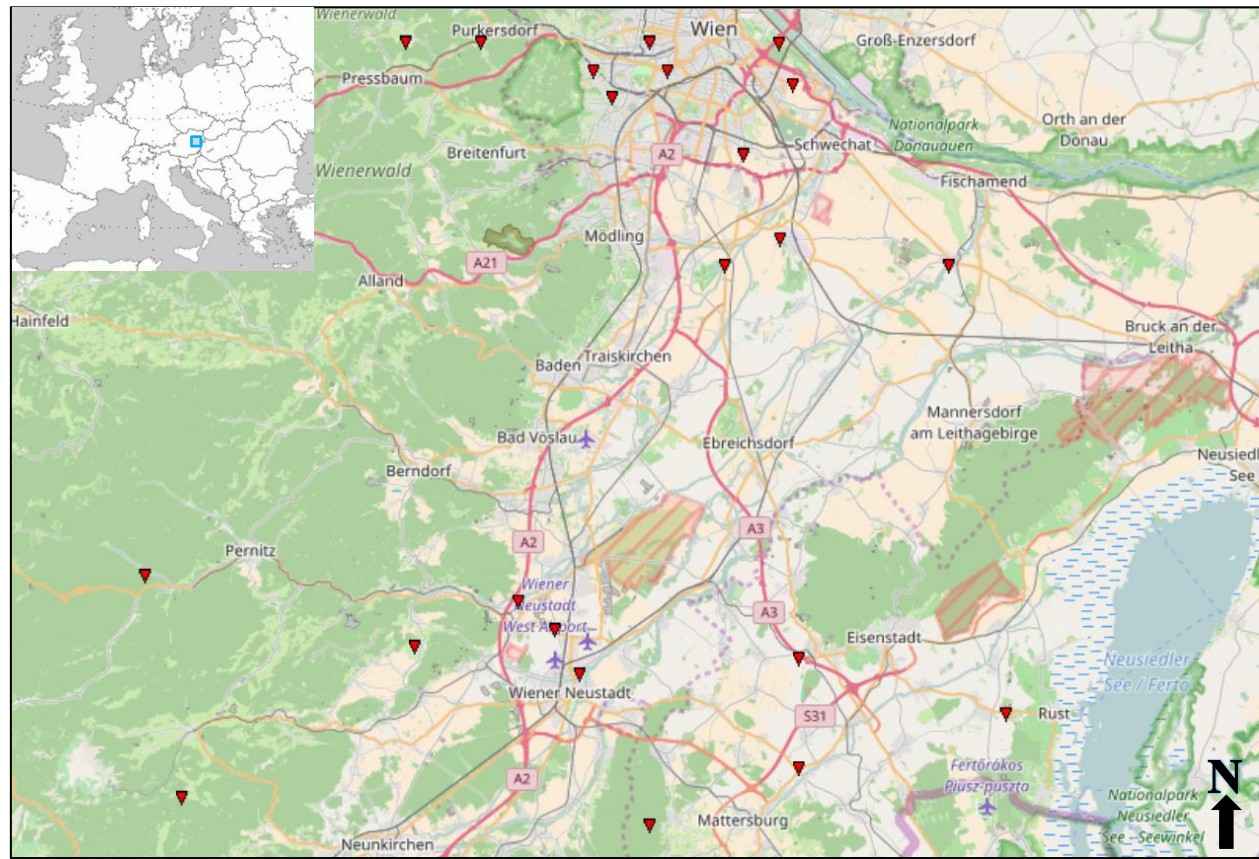

Fig. 1: Map of confirmed tornado events (ESWD QC1) in the natural environment of the southern Vienna basin around Wiener Neustadt (area of about 4 000 km²) from 1800 to 2016. Tornadoes are plotted as red triangles at the main damage site of an event. Source: www.eswd.eu as of 18 November 2016. Base map: www.openstreetmap.org.

Another motivation for in-depth documentation of high-end events is a need of real-world scenarios for the purposes of civil protection.

As a consequence the following aims arise:

10      a)   What kind of original sources can be found?

        b)   Was the catastrophic windstorm event a tornado?

        c)   How strong was the event on nowadays (tornado) damage rating scale?

        d)   Is it ultimately possible to draw a coherent picture of the entire event?



## 2 Methodology

### 2.1 Retrieval of historical sources and their preparation

Damage assessment of a recent tornado typically starts with a careful local inspection and site survey of the damaged area. Damage needs to be assessed as quickly as possible after the event struck, as clean-up is a major concern for this task. For

historical cases such an approach is not possible. Instead of field studies and site surveys a profound search for original sources and in-depth analyses of such needs to be performed, followed by a historico-critical evaluation of the image sources and text sources found.

In total 11 archives were browsed over a period of about two years, most of them non-digital paper archives like municipal record offices, local museums, and the Austrian National State and War Archive. The material found was sorted, classified

and digitalized. Original photographs were scanned and saved in high resolution.

All times stated in this work origin from historical text sources and refer to local time.

### 2.2 Geo-referencing of source information

The goal is to rate damaged objects and to locate them. Damage reports might mention specific details. Photographs from a certain perspective also reveal different details. Impressions of the damage and the sturdiness of the damaged object may

vary depending on the point of view, time of image taken, light conditions, and many other factors. In some cases more than one damage photograph is available for a specific object of interest. A careful geo-referencing of objects seen on a photograph is necessary in order to allow for the grouping of such photographs. This requires a forensic approach when it comes to the determination of the position of the photographer and his viewing direction. Some objects visible on the photographs may not qualify for a rating. Reasons can be that such objects are not clearly enough visible, or too far away, or

the part seen is not large enough or the sturdiness of the object cannot be determined. To summarize, not for all objects on a photograph there is enough information available to perform further backed damage rating. Many such objects can be classified as "damaged" (helping to determine the width of the damage path), but without a specific rating.

A typical precision of achieved object localization is on the order of distinguishing street numbers, i.e. one house from its neighbouring house, or between meters and deka-meters. In some cases historical street maps were necessary to perform this

geo-referencing task, especially where the building structures of whole blocks were modified and neighbourhoods rebuilt over time. In the given case, after carefully carrying out the mentioned steps of geo-referencing, it was possible to reproduce the path the photographer took 100 years ago for a number of the photos found.

### 2.3 Application of the DI-DoD approach to the historical material

Within ESSL, windstorm damage rating is performed with a damage indicator (DI) – degree of damage (DoD) – approach, based on the original Fujita Scale (Fujita, 1971) that was defined for a strong US frame house. In his memoirs (Fujita and




Merriam, 1992) Fujita refined the method to derive an F rating by introducing a conversion table for in total 5 other than strong US frame house DIs, in their nature generic building types: weak outbuilding, strong outbuilding, weak framehouse, strong framehouse, brick structure, concrete building. The conversion leads from a damage f scale to a windspeed F scale score by adding or subtracting pre-defined full scale steps, depending whether the building of question was stronger or

5 weaker than the strong US frame house. Fujita writes in his memoirs book: "For determining the F scale, we have to estimate the f-scale damage of any structure with a damage. Then we select the F scale as a combination of f scales and structure types."

Based on the Fujita conversion table from f to F, Feuerstein et al. (2011) proposed a decision matrix for damage assessments

that already includes the results of the conversion step and refines the outcomes to the lower or upper half of each original step. Already Fujita included vegetation damage for the rating - in the original version of 1971 in a broad descriptive form, in the memoirs book of 1992 in the form of a collection of cornfield damage pictures for every single F scale step from F0 to F5, called "F-scale damage chart" (parallel to one for the strong US framehouse). Feuerstein et al. in 2011 designed a vegetation decision matrix the same way as the building type matrix was set up, without including cornfield damage (Fig. 2).

The DI-DoD method is also being used by the EF-Scale (EF-Scale document, 2006) in the USA and other countries, but with more and more specific DIs and different degrees of damage. Outside of the USA many DIs cannot be used because of their non-existence that results from specific building codes and practices. In addition, DIs of the EF-Scale are not defined to be used for historical building types. That being said, the EF-Scale is hardly applicable for the re-analyses of historical

European tornado cases and was therefore not used in this study. Instead we used the Feuerstein et al. decision matrices.

| Fujita damage class | f0 | f1 | f2 | f3 | f4 | f5 |
|---|---|---|---|---|---|---|
| loss ratio (%) | 0.1 | 1 | 10 | 50 | 90 | 100 |
| degree of damage → / ↓ damage indicator | light roof damage | significant roof damage | roof gone | walls partly collapsed | largely blown down | blown away |
| A weakest outbuilding | F0+ | F0+ | F1- | F1- | F1+ | F2- |
| B outbuilding | F0+ | F1- | F1+ | F2- | F2+ | F3- |
| C strong outbuilding/ weak framehouse | F0+ | F1+ | F2- | F3- | F3+ | F4- |
| D weak brick structure/ strong framehouse | F1- | F1+ | F2+ | F3+ | F4- | F5 |
| E strong brick structure | F1- | F2- | F3- | F4- | F5 | F5 |
| F concrete building | F1- | F2+ | F3+ | F4+ | F5 | F5 |

| Fujita damage class | f0 | f1 | f2 | f3 | f4 | f5 |
|---|---|---|---|---|---|---|
| loss ratio (%) | 0.1 | 1 | 10 | 50 | 90 | 100 |
| damage prevalence → / ↓ damage indicator | extremely isolated | isolated | significant | frequent | prevalent | total |
| G branches - leafy | < F0 | F0+ | F1- | F1+ | F2- | F3- |
| H - bare | F0- | F1- | F1+ | F2- | F2- | F3- |
| I tree stands - diseased/ unstable | < F0 | F0- | F0+ | F0+ | F1- | F1- |
| J - strong | F0+ | F1- | F1+ | F1+ | F2- | F2- |
| K edge trees, hedges, underwood | F1- | F1+ | F2- | F2+ | F3- | F3- |

Fig. 2: Damage rating matrices for buildings (left) and vegetation (right), adapted after Feuerstein et al. (2011).



From each damage description or damage photo, those objects with usable information were classified, usable in the sense of clear-enough information for the DI and DoD. First, a most appropriate damage indicator was chosen, second the degree of damage was determined. Only the descriptive "damage prevalence" from the Feuerstein et al. matrix was used for this assignment, not the economic loss ratio. For each data point such an approach was taken, the outcome is a "first guess". In

case several damaged objects were visible on one image, each of them was rated independently. To make the outcome transparent for each usable damaged object, not only the final rating (for example "F3-") was documented, but also the underlying DI in letter abbreviation (for example "C" for a strong outbuilding or weak frame-house) and the determined DoD number from the small-f-class (in this example "f3" for "walls partly collapsed"). As it is evident, the information pair DI-DoD alone already contains all information, but as expressions in the Fujita scale were common worldwide in the past

and still are used in some parts of the world, we also present the outcome in the refined F-classes in order to allow for direct comparison with other datasets.

## 2.4 Aggregation of singular data points into damaged objects

After 2.3 a database of DI-DoD pairs from different information or image sources - mainly different photos (for example from various perspectives - needs to be condensed into a database of damaged objects. For each object only one damage rating shall remain, which is necessary in case of the availability of more than one singular data point. In this step all now available information for a given object can be taken into account. Additional hints can be used for the object rating, hints that may not have been present in the single piece of information in the first step - as a better view on the sturdiness of an

object or its construction practice and used materials. The more singular data points ("first guesses") are available as a basis for this step, the more confident one can be with the object rating. The outcome of this procedure is one aggregated rating for each damaged object.

## 2.5 Data mapping

For all now available condensed damage information their geographical position was plotted on a map. This work stage is

necessary to perform a plausibility check of local inhomogeneities along the damage path. In the case of this study a manual review of eye-catching outliers was performed.

Given the small scale phenomenon of tornadoes, temporal and spacial changes of windspeed in tornadoes can occur abruptly. Keeping this physical nature in mind, care needs to be taken not to overdo smoothing of tornado damage. This

iteration step shall mainly lead to another critical review of single suspect object ratings, not to a general smoothing. In case no additional reasoning was found to change the original rating (like a weakness in the construction of a wall, that was not recognized in earlier steps), the original rating was kept.



Only after this step the final object ratings are fixed and can be ultimately plotted and used for further analyses. In the case of the studied event, the density of final ratings was high for the urban area of the damage track (Fig. 3), while only single data points could be fixed for rural areas along the track to the West and East of Wiener Neustadt.

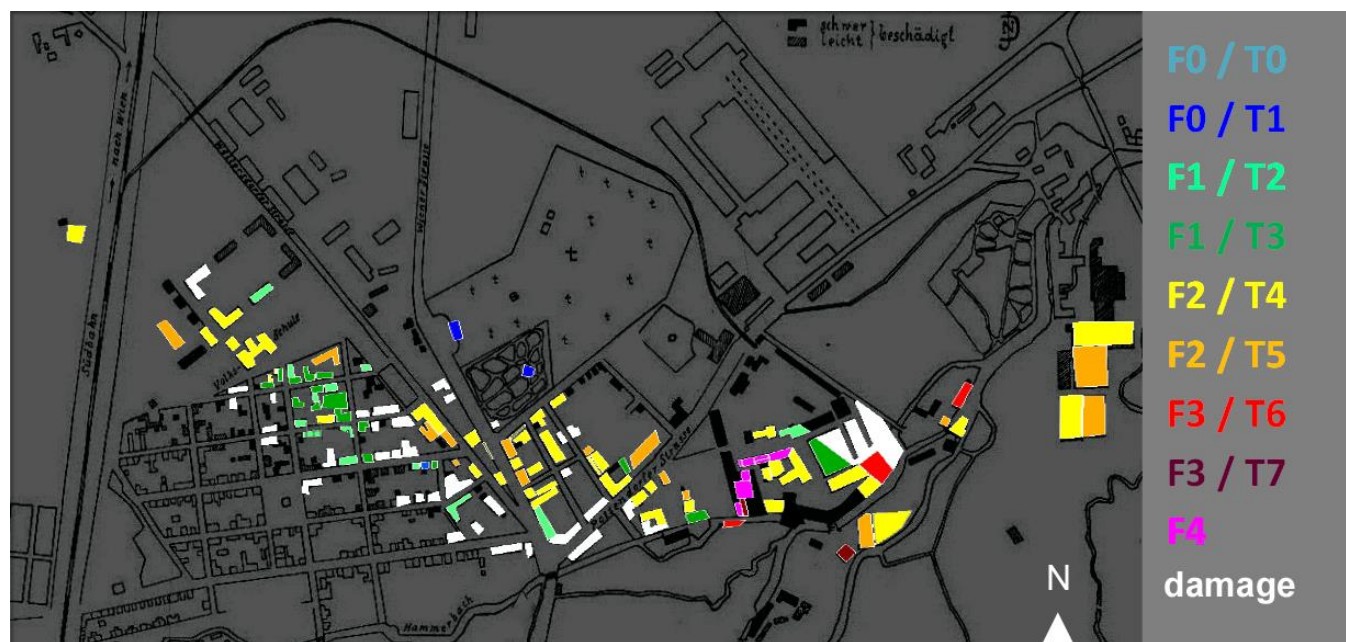

Fig. 3: Damage map of urban Wiener Neustadt area. Coloured and white shading represents new findings with corresponding scale values given to the right. The background map is a historic city map. Buildings shaded in dark grey have shown minor damage ("leicht beschädigt"), buildings shaded in black have shown major damage ("schwer beschädigt") in the historical damage survey.

In addition to the damage rating, information on treefall patterns, directional bendings of objects and falling directions of walls were documented, where available. It is important to keep all final data points and their DI-DoD pairs for future needs, not only a single final maximum rating for the whole event.

15 **3 Chronological event description**

**3.1 General weather situation**

The daily weather report from the Central Institute of Meteorology and Geodynamics (ZAMG, 1916) shows that on 10 July 1916 a shallow low was present over northwestern Europe, a weak high over the Southeast. Pressure changes were neglectable over eastern Austria the day before and after the event, while pressure was rising immediately after the





catastrophe. Temperatures were high in easternmost Austria, even for a mid-summer day. There was a temperature gradient present over central Europe from Northwest to Southeast, while the steepest gradient is present during the early afternoon just to the Northwest of Wiener Neustadt (Vienna Forest mountain range), see Fig. 4.

Fig. 4: Temperature map (degrees Celsius) of eastern Austria, 10 July 1916, 14:00, adapted with added geographical information after Dörr (1917)

Regarding moisture, dew points at 14:00 were measured 15 and 17 degrees Celsius respectively at different stations in Wiener Neustadt. In Vienna, some 50 kilometres to the north, the dew point was 18 degrees Celsius (ZAMG, 1916).

In the early afternoon weak winds from the Southeast to East were reported around Wiener Neustadt, while a strong northeasterly wind was reported about 100 km further north, and weak westerly wind was reported in the area 50 to 100 km to the West of Wiener Neustadt. Together with other weather station reports, it is evident that in the early afternoon a distinct





convergence zone was present in the surface wind field in the area just to the Northwest of Wiener Neustadt, mainly along the Vienna Forest mountain range. A mountain station about 50 km to the south in around 1000 m ASL reported strong southwesterly wind. On the Sonnblick Observatory, located on a mountain peak in the high Alps in 3106 m ASL further to the Southwest, gale-force winds out from the West-Southwest were recorded (Svabik, 2013). In the area northwest and

around Wiener Neustadt all weather stations reported thunderstorms during the afternoon and early evening hours, many of them together with gale-force wind gusts and hail. Immediately north of the windstorm damage track walnut sized hail was recorded (Dörr, 1917). In Wiener Neustadt alone thunderstorms were recorded in town from 16:45 to 17:30 and with strong gusts at 17:40, at the airfield a few kilometres to the northwest from 14:30 to 15:45, at 17:20 with gale force gusts, and from 17:30 to 17:50 together with hail. Also text sources confirm that there was a thunderstorm with gale-force gusts already

before the catastrophic event struck the town of Wiener Neustadt. After this storm of 17:20 it became calm again and very moist and sticky.

A reanalysis driven by the ECMWF model, ERA-20C (Poli et al., 2016) with horizontal resolution of around 125 km, allows for an investigation of large-scale conditions leading up to the tornado event. Reanalysis assimilates surface pressure and

marine wind observations. Poli et al (2016) suggest that realistic simulations of daily events are found for the regions and times, for which high coverage of observations is available. That seems to be the case at least in the studied region given number of stations mentioned above. As of 10 July 1916 15 UTC, reanalysis simulates a broad belt of strong, 20 m/s +, west-south-westerly flow at 500 hPa in the wake of a ridge stretching from Balkan peninsula towards Poland (Fig. 5a.). At 850 hPa, a wavy frontal boundary was located over eastern France, Germany and Poland (Fig. 5b.). Over eastern Austria, in the

warm sector and under weak southerly flow, temperature exceeded 16° C at this pressure level, suggesting the presence of very warm, tropical airmass. Near the surface, Wiener Neustadt was located on the north-western flank of shallow surface low with weak easterly flow at 10 m and in the moist airmass with dew point temperatures exceeding 16 °C. (Fig. 5c.). Note that simulated dew point temperatures correspond well with actual observations. Combination of these conditions created a setup favourable for severe, well-organised convection with CAPE values above 750 J/kg and 10 m to 500 hPa bulk wind

shear (or deep layer shear) exceeding 20 m/s (Fig. 5d.). Such degree of vertical wind shear is commonly associated with supercells (Thompson et al. 2003) and also strong tornadoes (Thompson et al. 2003, Púčik et al. 2015 or Taszarek et al. 2017). Supercells are virtually the only type of thunderstorms associated with violent, F4+, tornadoes (Smith et al. 2012). Despite seemingly conducive environment to severe thunderstorms, model resolution likely limits its representativeness of local conditions given the complex topography of the studied region.





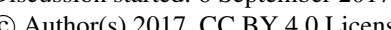

Fig. 5: Reanalysis of a. 500 hPa geopotential height (blue isolines) and wind (barbs and colour scale); b. 850 hPa geopotential height (blue isolines), temperature (colour scale) and wind (barbs); c. Mean sea level pressure (blue isolines), 2 m dewpoint temperature (colour scale) and 10 m wind (barbs); and d. CAPE (colour scale) and bulk vertical wind shear between 10 m and 500 hPa (barbs) on 10 July 1916 15 UTC. Location of tornado is indicated by red triangle in each of the panels.

## 3.2 The tornado(es)

Out of both the findings of the scientific paper that was written after the event (Dörr, 1917) and out of four eyewitness reports (Karl Bauer from the village of Lichtenwörth, Dr. Alfons Friedel from the village of Wöllersdorf, Prof. Leopold



Schmidt in a train from Leobersdorf to Wiener Neustadt and a Lieutenant of the Austro-Hungarian army in a train from Wiener Neustadt to Vienna) it becomes clear that the nature of the wind event was a tornado. Descriptions included the depiction of a funnel or column and compared the visible notion of the tornado with smoke-like appearance. The German terms for a whirlwind were used repeatedly in contemporary texts. Also the long but narrow track is a strong indicator for a

tornado, in addition to the specific falling direction of objects relative to their position to the estimated centre of the storm.

Exact time readings of the beginning and end of the event do not exist. The maximum time range for the catastrophe can be narrowed down between 17:30 and 18:00. An event duration for the whole track of only 10 min mentioned in the "Meteorologische Zeitschrift" (Dörr, 1917), without quoting reference times, seems inappropriately short. In this work for a

track length of about 1 km a time span of already 4 min was quoted, leading to inconsistent estimates for the total track.

An early anemograph - a model of it still to be viewed in the city museum of Wiener Neustadt - recorded a wind speed of "above 40 m/s" (Dörr, 1917) on the western airfield of Wiener Neustadt over a duration of 3 min. Given a lack of precise time readings, the translational speed of the tornado cannot be estimated with confidence. Two independent eye witness reports exist for the time immediately before the main tornado struck the town of Wiener Neustadt. Both reports are credible

and are in line with documented damage, but - and this is highly relevant - describe different tornadoes. We therefore here use the term "predecessor tornadoes". While in the "Meteorologische Zeitschrift" only a western predecessor tornado is mentioned, another eye witness report (WNN, 1916) describes only the existence of a northern ancestor. This leads to the scenarios that in the area of the western airfield either two predecessor tornadoes merged into the main tornado, or that one of these tornadoes became dominant while the other dissipated closely. Without knowing precise time readings, the

following chronological order can be reconstructed.

### 3.2.1 Western predecessor and main tornado

In the village of Peisching, located in the Piesting valley, the first damages were documented (Dörr, 1917), some 15 km west-north-west of Wiener Neustadt (Fig. 6). The tornado was reported to have formed on a slope to the west of this village,

while in Peisching roofs were damaged, and a continuous damage track was visible from here onwards to the river Leitha east of Wiener Neustadt. In the northern part of the following village Dreistetten a few houses were severely damaged, and the first severe injury is evident (Karl, 2017; Schramböck, 2017). A young farmer, who was at home on furlough from World War I, was picked up by the tornado a few metres in height, and thrown back to the ground. As a result of his severe leg injury he could not rejoin the battle – a rare case of possible prevention of a war fatality by a tornado.

The further track over the hills of the Mahlleitenberg was marked by massive forest damage. On the eastern slope the tornado descended 200 m of height down to the "Steinfeld" plain in the southernmost part of the Vienna basin within only about two kilometres of horizontal distance, with further direction towards the western airfield of Wiener Neustadt. The width of the damage path is unknown up to this point, and no detailed assessment of the forest damage is recorded. As of local personal communication (Karl, 2017; Schramböck, 2017) there was no continuous track through this large forest area.





But some patches of forest were severely damaged along the imaginary line between the tornado damage in Dreistetten and the appearance of the tornado west of Wiener Neustadt, for example at a place called "Fiedelwiese". Several wooden rafter beams from a house in Dreistetten (called "Tilt'sche Villa", Haltergasse 93) were carried by the tornado over a distance of about 3 km and stuck there inclined into the soil of a forest clearing meadow east of Dreistetten (called "Zwara-Wiesn" or
"Hinterwiesn").

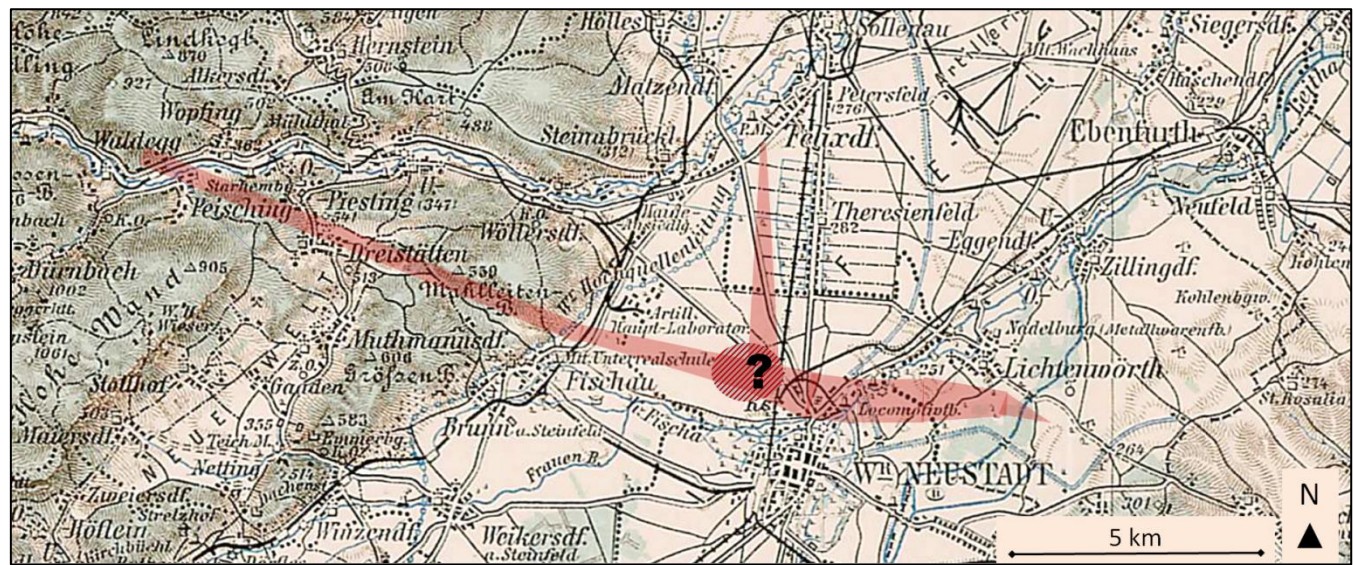

Fig. 6: Map of tornado damage tracks (red shadings) in the Wiener Neustadt area. The hatched area with the question mark labels the airfield west of town, where the exact genesis of the main tornado is unknown (merger or dissipation of one of the
predecessor tornadoes). Basis map: „Generalkarte von Mitteleuropa des k.u.k. Militärgeographischen Instituts Wien, Blatt 34° 48° Wien" of 1893.

Leaving it open whether the tornado merged here with the second predecessor tornado or not, the main tornado entered into the urban area of Wiener Neustadt just after passing the major railway lines between the cities of Vienna and Graz. Airfield
staff reported that the tornado noticeable changed its moving direction over the airfield from east-south-east to south-east. This could be an indicator of the interaction with the northern predecessor tornado, while the northern tornado itself seemingly was not discernible to the airfield staff. Reasons could be rain shafts, lighting conditions, or size differences, to name a few.

After the passage of the railway tracks, the damage path is clearly verifiable, because of a high density of damaged objects. Over the northern neighbourhoods of the town, called Josefstadt, the tornado performed another turn and from here on moves straight to the east, after the main impact to the old locomotive factory even to east-north-east. After crossing the



northern parts of Wiener Neustadt, the tornado moved over open fields and dissipated soon after crossing the small river Leitha, some 5 km to the east of town.

### 3.2.2 Probable northern predecessor tornado

5 The eyewitness report of Professor Leopold Schmidt (WNN, 1916) suggest the existence of a second predecessor tornado. Schmidt was an aviation pioneer at the local airfield and travelled at the time of the tornado event in a train to Wiener Neustadt. At 17:28 Schmidt departed in the village of Leobersdorf, located 13 km north of Wiener Neustadt. He observed an intense thunderstorm over the Triesting valley. 2 km west of the village Felixdorf Schmidt observed a "small whirlwind" that moved with similar speed and direction as the train to the south. Close to Wiener Neustadt the size of the tornado and its 10 translational speed increased. Dark farmland soil coloured the whirlwind column deeply black with a distinct outer boundary. The large funnel moved towards the airfield hangars.

Interestingly, this trustworthy eyewitness report, although published soon after the event, was not mentioned in the scientific study of the "Meteorologische Zeitschrift" (Dörr, 1917), the more being in contradiction to the conclusions of the Dörr 15 study. Schmidt was co-founder of the airfield and lead wind measurements with a self-constructed anemometer. He was very familiar with the landscape around his airfield, knew the distances and observed several minor whirlwinds before, as we learn from his report. We therefore assume that Schmidt would have been able to distinguish the western predecessor tornado that came down from the forested hills from the phenomenon that he describes in his report from the northern train ride. Again, it can only be speculated why he did not discern the other, western, tornado. Given the clear description that 20 perfectly fits for a tornado, downbursts or a brief gustnado would be unlikely options.
Independent from the findings of the eyewitness reports, our analyses of the damages reveal that two objects well north of the western predecessor tornado were indeed heavily damaged. This fits well with the probable northern tornado track.

Both predecessor tornadoes must have co-existed for at least several minutes and must have merged or approached each 25 other in the area of the airfield in a very narrow time span.





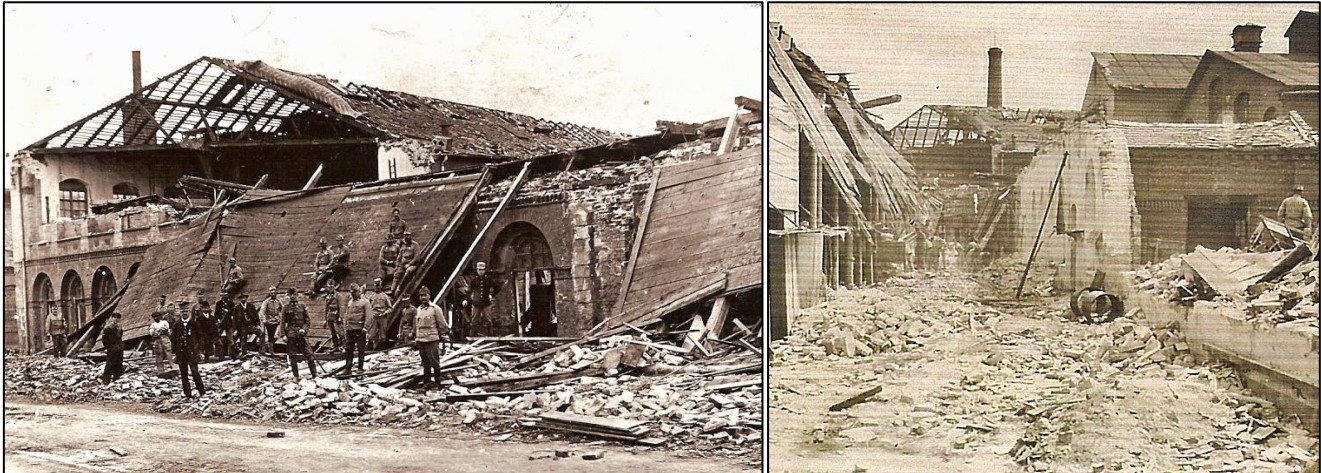

Fig. 7: Details of the old locomotive factory. Left: inner ceiling flipped outside, in foreground whole upper floor with massive brick walls collapsed (similar as the building parts that are still visible in the background). Right: Backyard of the same building with destruction of up to 1 m thick walls down to the ground level. Source of photos: Industrieviertelmuseum

5  Wiener Neustadt.

### 3.2.3 Tornado impact in town and dissipation

Over the Josefstadt neighbourhood a maximum intensity in the upper F2 range of the Fujita Scale was reached, while the damage width of F0 measures about 400 m. An eyewitness reported a visible diameter of the tornado of up to 1000 m

10  (WNN, 1916). Further east, in the area of the old locomotive factory, the maximum intensity reached F4 (Fig. 7 to Fig. 9). The width of the damage track at this most devastated place measures about 600 m.





Fig. 8: F4 destruction on the plot of the old locomotive factory. Photo committed from a private archive in Wiener Neustadt.

As can be seen from the aerial photograph (Fig. 9), the degree of damage varies substantially within extremely short ranges

5    from nearly untouched to total destruction. This substantiates a statement by Fujita and Merriam (1992): „These examples

show that tornado winds are highly variable within a short distance both in wind direction and speed, necessitating the use of

all types of structures in mapping the F-scale wind pattern.“





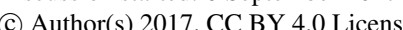

Fig. 9: Aerial view of parts of the most damaged old locomotive factory. Most destructed area in the lower right quarter of the photo. Source: Industrieviertelmuseum Wiener Neustadt.

The main support for the F4 rating in the most devastated area is the fact that several brick walls of about 1 m thickness belonging to the old locomotive factory collapsed. The ceiling between the ground floor and the first upper floor of this building was supported by steel beams, leading to an even more robust sturdiness. A large piece of this floor was flipped outside the exterior walls. An aggressive rating could have called the DoD "largely blown down" for important parts of the factory in an area of about 50 m of diameter, which would have resulted in a final F5 rating. Given the uncertainties in the

state of conservation of the building and the in detail unknown building structure before the event, we decided for the "walls partly collapsed" DoD, resulting in the final F4 rating. A neighbouring building also experienced the collapse of very thick brick walls, resulting in another F4 rating.

Further east, at the Daimler factory, the maximum intensity decreased to the upper F2 range, and the damage width

decreased to 400 m. After leaving the town, the tornado passed over grassland and crops and reached the river Leitha and its





surrounding forest to the south of the village Lichtenwörth, measuring only 80 m in diameter. Here the last trees were downed and the tornado dissipated soon after.

## 4 Results and new findings compared to the historical study

### 4.1 Facts and figures

a) The manual search in the different archives unveiled more than 100 damage photographs of the tornado event. In total 180 single damage sources were found.

b) Shortly before the "Tornado of Wiener Neustadt" struck the town, two predecessor tornadoes existed.

c) The maximum intensity of F4 is supported by the collapse of thick brick walls of two neighbouring buildings. The

heaviest damage was found in the old locomotive factory, where brick walls of about 1 m thickness were levelled.

d) The tornado path including the western predecessor tornado had a length of 20 km.

e) The maximum width of the F0 damage was 600 m, the visible diameter was estimated to be 1 km.

f) The area under the influence of at least F2 winds comprised 20 ha within the city of Wiener Neustadt, the area under influence of at least F0 winds 81.5 ha in the urban area. More than 100 houses were affected in the town.

15 g) The final number of fatalities within the urban area of Wiener Neustadt was at least 34.

### 4.2 New findings

At the time of the event no conceptual model of tornado formation was known, but the co-existence of severe thunderstorms
was noticed. Historical weather data together with the eyewitness reports reveal that a typical setup for severe convective storms was present. Obviously a distinct convergence zone, evident from the historical weather station readings, provided the necessary mesoscale lift to initiate the thunderstorms. Vertical wind-shear was present, as historical mountain observatory weather reports suggest, and high moisture according to the dew point readings and descriptions of a muggy afternoon. Instability must have been high, because of widespread intense thunderstorm activity in the region together with
reports of large hail. This is in accordance with the modern theory of favourable environmental conditions for the development of organized deep convection and tornado genesis (Church et al., 1993).

Out of the thunderstorm reports from the dense weather station network around the damage site, it can be said that the thunderstorm activity was not only widespread and long-lived, but also complex with stations being struck several times.
The detailed interactions of the different cells cannot be reconstructed without modern remote sensing techniques or photographic documentation of the storm developments.



During the time of tornado activity there must have been several cells active in the immediate vicinity of Wiener Neustadt. From the description of large hail and the strength of the tornado, a supercell can be taken as granted. The remarkable complexity can be seen as a prerequisite for the uncommon development of the two plausible predecessor tornadoes of the main tornado that hit Wiener Neustadt (Fujita and Merriam, 1992).

The existence of two predecessor tornadoes is a compulsive deduction from the analyses of different trustworthy sources and a new hypothesis in contrast to the contemporary scientific study (Dörr, 1917). Today we have no means to find out whether an eyewitness report speaking about a northern predecessor tornado was unknown to Dörr or if it was knowingly ignored. The professional background of the eyewitness in question, Professor Leopold Schmidt, is fascinating: Schmidt was one of

the founding fathers of the airfield in Wiener Neustadt and an aviation pioneer (ÖBL, 1992). He conducted wind measurements by a self-constructed anemometer, which he statistically analysed. Schmidt was familiar with the landscape around the airfield and was therefore one of the best persons to estimate distances and correctly observe meteorological phenomena. In the years before the event he observed a number of lesser whirlwinds in the area, therefore he also was able to distinguish a tornado from this different fair weather phenomenon. To summarize, we assume that Schmidt would have

been able to correctly position and identify the western predecessor tornado. But, instead, Schmidt transmits us a report about another tornado that happened about the same time but further north and with a different moving direction. For unknown reason the different eyewitnesses were not able to see "the other" tornado respectively in the minutes before the big tornado finally hit Wiener Neustadt. An idea of how complex tracks of different tornadoes in one close area can be we can get from a track map that was made after the 3 July 1980 Grand Island, Nebraska, tornadoes (p. 44, Fujita and Merriam,

1992), showing between the tornadoes no. 5 and no. 6 a similar angle as can be seen from our Fig. 6.

This work documents a rare occasion where a tornado descended 200 m of height within a horizontal distance of 1.7 to 2.9 km, depending on the exact track. The western predecessor tornado crossed a small mountain range before it reached the plain surrounding Wiener Neustadt. Again, a similar case can be found in a tornado track map of Ted Fujita on the Sayler

Park, Ohio, tornado of 3 April 1974 (p. 287, Fujita and Merriam, 1992), where a tornado descended about 350 m of height within a horizontal distance of 2.5 km.



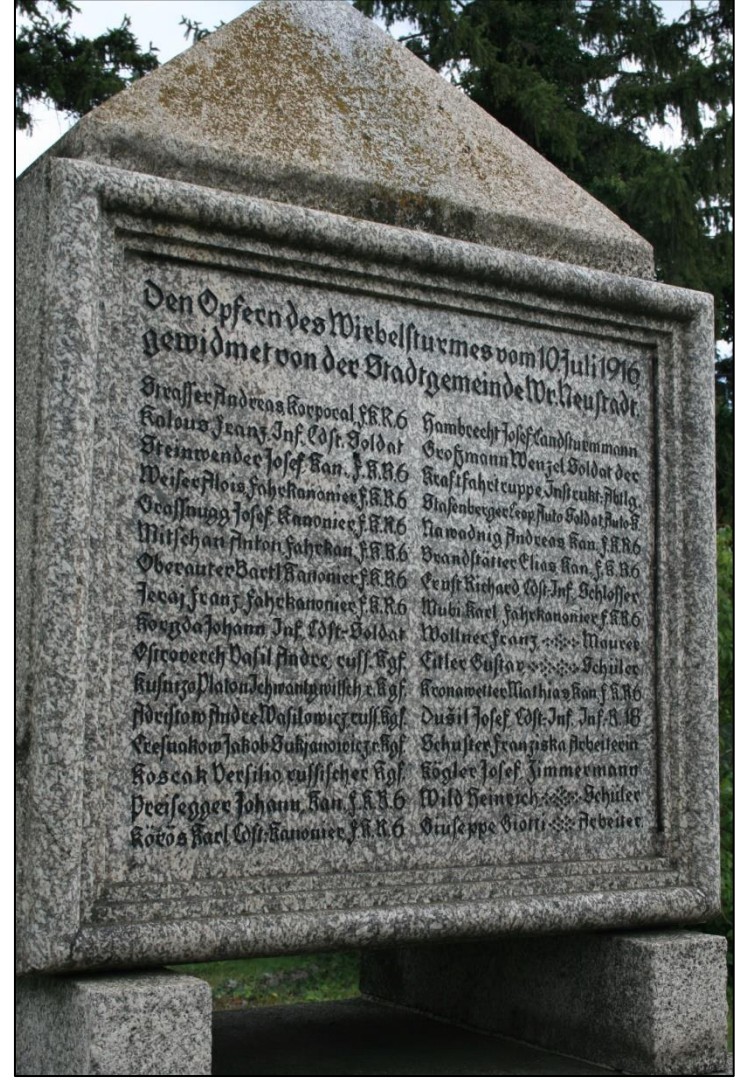

Fig. 10: Memorial stone on the cemetery of Wiener Neustadt, erected above the common grave with 32 of the tornado victims. Text: "Den Opfern des Wirbelsturmes vom 10. Juli 1916 gewidmet von der Stadtgemeinde Wr. Neustadt [Namens- und Berufsliste]"- Translation: "Dedicated to the victims of the whirlwind of 10 July 1916 by the City of Wiener Neustadt [list of names and affiliations]. Photo: Alois M. Holzer.

By applying some kind of forensic work an error in the originally stated total number of fatalities was revealed. All historical sources quote a number of 32 fatalities. Also the official police list published on 15 July 1916 (WNN, 1916) and the remembrance stone on the cemetery (Fig. 10) list 32 persons. The police list also names people who were severely injured. Some of these names later appear on the remembrance stone on the cemetery. It is plausible that these persons died soon after the event due to their severe injuries and were buried together with most of the other victims in a common grave on the



cemetery of Wiener Neustadt, where the memorial stone was erected later on. A side note within one of the eyewitness reports (Bauer, 1916) tells us that a few victims were buried in neighbouring villages. These names that originally appear on the police list finally are missing on the stone. While unnamed persons on both lists could slightly further increase the number, no fatalities outside of Wiener Neustadt are known. To summarize, this brings the total number of fatalities in town

to at least 34.

**5 Implications and Outlook**

Finding 4.1 a) implies that it can be rewarding to search historical archives for damage photos and reports of possible tornado events, and severe storm events in general. In this case the number of damage photos found was a multiple of the expected number. The archives of Wiener Neustadt even survived massive bombings in the Second World War.

The authors of the study found that the rating on the Fujita Scale into half-steps above F3 would have resulted in an over-confidence of the rating precision. The authors therefore refrained from it. It should be left open if such precision can be achieved in other cases.

In some cases the assignment of appropriate DIs was difficult, not because of a lack of information, but for reason of a lack of selectable DIs. Therefore a wider range of DIs would be desirable both for buildings and vegetation.

Regarding vegetation, already Fujita (pp. 33 and 283, Fujita and Merriam, 1992) used damage signatures in cornfields and in forests alone (without building references) for classifications up to F5. Fujita documented different stages of vegetation

damage on a series of photos for the whole range of the F-Scale. Additional damage indicators for vegetation were recently presented by Hubrig (2015) in a detailed table and can be used in future. These DIs for mainly wooden plants were not yet available for this study.

At the time of completion of this paper an initiative rose to harmonize tornado ratings worldwide. An international working

group formed with the intent to assess the current status of tornado ratings globally and the final goal to reach consensus on an International Fujita Scale (IF-Scale), as proposed earlier by the European Severe Storms Laboratory (Holzer and Groenemeijer, 2015). Harmonization of tornado ratings has a wide range of implications, from applications in the engineering sector to climatology, and will facilitate future tornado research.

Such a harmonized tornado scale can only be a first step towards a more reliable estimation of tornado wind speeds, as it still heavily relies on expert guesses, especially in the upper range of the scale. More research is desirable in the form of experiments, calculations and observations, to support better estimates in the case of a given damage.



It has proven useful to document a historical tornado case in great detail also for the purpose of civil protection and citizen science. The given outcome served as a scenario for civil protection exercises in the town of Wiener Neustadt. On the occasion of the 100 year recurrence of the tornado event an exhibition was opened in the City Museum of Wiener Neustadt

that became one of the most-visited exhibitions of the year. In addition guided tours along the damage path were organized by ESSL and raised significant public interest. Historical damage photograph posters were presented and explained on site of the original damage.

## 6 Summary

The herewith documented working method can be exemplary for the analyses of similar cases by closely following these

steps in chronological order:

- Retrieval of historical sources and their preparation
- Geo-referencing of source information
- Application of the DI-DoD approach to the historical material
- Aggregation of singular data points into damaged objects

- Data mapping

By application of this method the earlier raised questions could be answered:

a) A surprisingly high number of both damage photographs and damage reports was found in historical archives.

b) The catastrophic windstorm event could clearly be classified as a tornado.

c) The tornado after 100 years could be rated as F4 on the Fujita Scale.

d) The weather situation was typical for a severe weather outbreak. Along uninhabited parts of the damage track open questions remain, especially regarding the formation of the main tornado that finally struck the town of Wiener Neustadt. In the urban area the damage path was clear and due to a high number of damage indicators worth to study in detail. The big tornado that struck Wiener Neustadt formed out of two predecessor tornadoes that

approached each other in an angle of about 70 degrees, representing a rare case of complex tornadogenesis.

The tornado was rated F4 with a total length of the damage track of 20 km, including the western predecessor tornado, and a maximum tornado width of 1 km. The number of fatalities was raised from the number of 32 in the original sources to at least 34 according to own forensic findings.


To facilitate such work in future, more DIs for buildings and for vegetation are desirable. ESSL participates in an international working group with the goal to reach consensus on an International Fujita Scale (IF-Scale).



**Acknowledgements**

We thank Mathias Stampfl for the initiative for this project, for his practical effort during the project work, for the essential help in the accomplishment of the guided city tours along the historical tornado track, and for other voluntary work that was crucial for the success of this undertaking.

We acknowledge support from Pieter Groenemeijer and Georg Pistotnik, who provided input to the project report. Charles A. Doswell III provided us with independent ratings for selected damage photographs, providing his experienced US-view on some of the historical but crucial damage sites.

We like to thank Otto Svabik, now retired from ZAMG, for the provision of the original historical weather observations from Austrian mountain observatories.

We thank the ESSL-internal and external reviewers of this paper for their helpful feedback. Without the large number of people helping us with tips, hidden information or other support to reach our goals, this project would have failed. We are therefore grateful for their attitude.

This work was supported by the City of Wiener Neustadt under the project title "TORNeustadt".

The following archives provided us with original sources, which we are thankful for:

Industrieviertelmuseum, Stadtarchiv Wiener Neustadt, Stadtmuseum Wiener Neustadt, Österreichisches Staats- und Kriegsarchiv Wien, Freiwillige Feuerwehr Wiener Neustadt, and several private persons.

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
