# Peer review of "A forensic re-analysis of one of the deadliest European tornadoes"

_Natural Hazards and Earth System Sciences, 2017_

## Referee Comment (RC1) · Anonymous Referee #1 · 16 Sep 2017

Review of "Re-Analysis of one of the deadliest Tornadoes in European History and its implications"

I want to like this paper a lot and recommend publication. The analysis of the damage classification was obviously thorough, and the authors are well-intentioned by compiling these various sources into a single database for analysis. Unfortunately, the paper falls well below expectations. I found the text extremely difficult to follow.

In my view, the current poor state of the manuscript shows a disregard for the volunteer time of the reviewers who are forced to try to make sense and provide detailed comments to improve a manuscript. Such revisions are the responsibility of the authors, not the reviewers. In other words, it is not up to us to clean up your mess. Sorry for being so frank, but I am not pleased with the state of this manuscript.

[Figure]

A paper should make it clear to the reader what its goal is. That is nowhere clear. The abstract vaguely talks about "demonstrating feasibility", but it is not clear of what. This paper needs a clear single-sentence statement: "The purpose of this paper is to...". If the authors can't do this, then there is little justification for this paper.

After the text is made more clear, the paper could use a thorough proofreading by a native English speaker to make it grammatically correct.

Finally, I will add that the paper was made even more difficult to read because the text lacked indented paragraphs or vertical spaces between paragraphs.

Recommendation: Reject and resubmit after substantial revision, rewriting, and proofreading.

Abstract:

Lines 8-10: The first sentence of an abstract should be clear. This is difficult to read and understand. What are the authors trying to say?

"Widen the data basis"? What does this mean?

"It could be speculated": By whom?

Abstracts should be one paragraph.

Rather than saying "After presenting the methodology", the authors should describe the methods and data.

"complex thunderstorm activity in the study area": This description is inadequate as a sound meteorological description of the event. "Complex" says nothing meaningful.

Untypical should be atypical.

"In the outlook": Outlook of what? The reader has no context for what this means. Clearer writing is needed.

Rather than say "we stress the side-benefits of the given study", say explicitly what they

are. A reader might not bother reading the entire article, so you should leave them with the most important results in the abstract. On this and other topics.

Introduction:

The introduction is poorly organized. It's a list of thoughts put together with no coherence and no argument that it tries to advance.

Line 24: "Before this study, on a regional level, and in scientific literature also on a national level, it was well known that". This content is entirely unnecessary. It is verbose, wastes readers' time, and does not encourage people to read further.

First paragraph: Starts out talking about the event of 1916, but then continues to talk about the terminology. A paragraph should have one consistent topic and theme. The reader has no context for the windstorm of 1916. Develop that topic first, then discuss what the terminology means.

Page 2: Lines 21-24: Why is this connected to the text before or after? This paragraph just hangs there. So what does the reader need to know about the surrounding data for the tornadoes? Why does this relate to the 1916 event?

Page 3: Paragraph starting at line 6 is just one sentence. This is improper in a scientific journal article.

I got frustrated in section 2 with the poor quality of the writing and lack of organization. As such, more detailed comments have not been provided. I am happy to consider providing more thorough comments ONLY AFTER SUBSTANTIAL REVISIONS FOR READABILITY HAVE BEEN PERFORMED.

Other:

1. Why is an international Fujita scale needed? What is wrong with the existing system that a simple modification to account for other damages be added?

2. What is the DI-DoD approach? Is this unique terminology and an approach used by

this study? Or, has it been employed previously? More needs to be said about this. In its present state, it is just introduced and the reader is assumed to know what it means.

3. P. 5, line 5: Specific page numbers should be provided with direct quotations.

4. There appear to be only two new figures (Figs. 3 and 5), but each have a different purpose. Is the purpose of the paper to re-examine damage classifications (Fig. 3)? Or, is it to describe the meteorology (Fig. 5)?

5. I found the general meteorological description in section 3 lacking. What is its purpose? I don't find that it is necessary to this paper. If I am wrong, then the authors need to make more clear why it is important. Again, this issue relates to the purpose of the paper, which is poorly stated, if at all.

6. Reference is made repeatedly to "thunderstorms", but I don't see any reference to cloud-to-ground lightning reported by observers. Are these really thunderstorms, or just convective storms (no lightning or thunder observed)?

7. References to "modern theory" citing a single paper from 1993 is inappropriate. In fact, the whole manuscript is lacking the most relevant and recent research on convective storms and tornadogenesis. Specifically, not a single paper by Markowski has been cited.

8. Section 5 is poorly motivated and lacks a coherent argument for the needs for these tools and approaches. Consider the types of papers written by Carles Doswell that show depth of insight and clarity in thinking about the need for European forecast improvements. Such clarity should be mimicked in the present paper, if the authors wish to motivate the European audience to act.

9. When I get to the end of the paper, I don't see the take-home message.

10. Acknowledgements should be one paragraph, not separate sentences.

References:

These two references to papers from ESSL have been omitted and should be considered to be cited in the revised manuscript.

Groenemeijer, P., T. Púčik, A. M. Holzer, B. Antonescu, K. Riemann-Campe, D. M. Schultz, T. Kühne, B. Feuerstein, H. E. Brooks, C. A Doswell, H.-J. Koppert, and R. Sausen, 2017: Severe convective storms in Europe: Ten years of research and education at the European Severe Storms Laboratory. Bull. Amer. Meteor. Soc., doi: 10.1175/BAMS-D-16-0067.1.

Antonescu, B., D. M. Schultz, A. Holzer, and P. Groenemeijer, 2017: The risk of tornadoes in Europe. Bull. Amer. Meteor. Soc., 98, 713–728, doi: 10.1175/BAMS-D-16-0171.1.

---

## Referee Comment (RC2) · G. Kopp (Referee) · 6 Oct 2017

General Comments This is a well-written paper which develops a coherent methodology for assessing tornado intensity and track details from historical sources, and then applying the method to a historically significant event. All tornado archives use re-analysis of historical events, so clarifying the methodology is of importance for users of such databases, as is data storage for future users. In addition, from an international perspective, developing a clear description of what tornado damage for the most severe European tornadoes looks like is important, since it will contrast with the distinctly different damage in the United States and elsewhere, where building practices and styles differ (e.g., wood-frame houses are the primary indicator for severe tornadoes in North America where the Fujita Scale originated). Such work will ultimately be

important for establishing a unified and consistent international standard for tornado intensity estimation.

Scientific Questions There is significant uncertainty with the authors' analysis, which I believe they have addressed with reasonable effectiveness. However, there are a couple of ways in which the analysis could be extended to reduce the uncertainty. The F4 rating of the tornado is centred on a brick building with 1m thick walls. It is not clear if this was typical practice, then or now, but one wonders if the capacity of such a wall system could be estimated from current engineering practice or literature. Such an analysis could ultimately provide support for the current rating (or a different one), when combined with appropriate aerodynamic data. The aerodynamics of a wall system, after the roof has been removed, are straightforward and unlikely to be altered much by details of the tornado vortex structure and wind field. I am not suggesting that the authors have to conduct this analysis; however, this aspect of tornado-intensity estimation is not mentioned in manuscript even though it is useful and becoming common amongst engineering analyses of tornado damage. A second technical aspect, which is typically important in severe tornadoes is wind-borne debris. I wonder, particularly, about the effects on trees. In severe tornadoes, one typically sees trees that are shredded by the debris (at least in North America), but Figure 9 does not appear to indicate that. Once wonders if observations are available, but are just not reported by the authors.

Technical Corrections • P.2, line 10. Building aerodynamics and structural analysis are much further advanced nowadays as well. • P.4, lines 20-21. Unclear/awkwardly-worded sentence. • P.4, line 30 and following. One could argue that the DI/DoD approach arose with the EF-Scale, not the original Fujita Scale, although the authors are treating the DIs in a simple way that is perhaps more consistent with the Fujita-Scale than that currently used in the EF-Scale. A sentence or two on this would be helpful. • P.6, lines 1-2. Unclear/awkwardly-worded sentence. • P.6, line 3. A sentence or two about the meaning and interpretation of "damage

prevalence" would be helpful. • P.7, Figure 3. It would be helpful to have the track boundaries identified on this map. • P.18, the sentences around line 10. No need to repeat the text from earlier in the manuscript.

---

## Short Comment (SC1) · 2 Nov 2017

This is not a full review but a list of comments which, I hope, are helpful for the authors in completing their study.

(1) A comment on additional information not yet included:

Contemporary newspaper reports uniformly state that 39 fatalities of the tornado were buried at the cemetery of Wiener Neustadt two days after the tragedy. This number is remarkably consistent in all newspapers, though many of them had sent their own correspondents to the site of the tragedy. It can therefore be assumed that this number also had an official origin (possibly "Korrespondenz Wilhelm", then the official organ of the empire, though this was not mentioned anywhere). There is no obvious explanation

for the discrepancy between this piece of information and the "official" police list with 32 fatalities referred to in this paper, but at least the possibility should be taken into account that the police list could have been incomplete.

Most notably, the list on the memorial stone depicted in Fig. 10 - if I understand it correctly that it is identical with the police list - contains mostly German and a few Slavic names, reflecting the blend of population near the capital of the Habsburg empire, a melting pot between "indigenous" people and migrants attracted from its Slavic crown lands. However, contemporary sources state that a large number of workers in the locomotive fabric, and presumably other places nearby, were Russian prisoners of war. Given that a considerable number of Russian citizens must have been exposed to the tornado when it reached its peak intensity, it would be highly surprising if none of them had fallen victim to its furor. The IMO most plausible explanation is that the Russian victims of the tornado were also among the 39 people buried during the solemn funeral, but stayed anonymous in this process and, in particular, were not filed by the police. There is likely no way to prove this assumption right or wrong anymore, but I think some considerations like these should be addressed.

In addition, contemporary newspaper reports also include at least half a dozen other detailed eye-witness reports, which were not quoted in this study but contain valuable information about the tornado's track, width, damage marks, and other characteristics. In particular, there are two reports with detailed information from Dreistetten, the village where the tornado track started. They also include an additional victim, a 12-year old boy lifed up and killed by the tornado in Dreistetten, which is not mentioned in this manuscript yet.

These yet lacking information sources may seem minor in comparison to the huge amount of de-archiving the authors have already done (which I explicitly appreciate!). However, this study is still incomplete as long as they are not included. Playing the devil's advocate, one could state that the authors already fail when addressing the first of their research questions (p. 3, line 10): "What kind of original sources can be found?"

All of these newspaper reports are readily accessibe at the Austrian National Library. The additional amount of work required to browse them and to process their information is limited.

(2) A comment on the meteorological reconstruction:

While I can follow the authors' main conclusions in section 3, it is a bit disappointing to see them interpret the surface wind and pressure field of a 125 km spaced reanalysis (without me wishing to dispute the big value of ERA-20C). This does not live up to their claim of a "forensic" approach which they state several times in the manuscript. It is obvious that not every detail can be reconstructed, not every hypothesis proven 101 years later; this is nothing to be ashamed of. However, weather reports from eye-witnesses, newspapers, and historic weather maps would allow a much closer reconstruction and a much stronger emphasis of potentially important aspects and processes. I am most intrigued by the sharpness of the thermal boundary, shown in Fig. 4. Information on the station height, not given in the manuscript (e.g., one of the 33°C measurements was taken at 850m), reveals that the thermal gradient amounted to >10K across 100 km in terms of potential temperature. It is extremely rare for a pre-convective situation in European mid-summer to feature such a strong thermal contrast. It is unlikely that a synoptic frontal zone alone could have created and sustained it against the influence of strong diurnal heating, which tends to level such differences. Instead, a true forensic analysis should place more emphasis on the "footprints" of previous deep convection prior to the tornadic storm.

Weather and newspaper reports mention that strong thunderstorms with heavy rain and very high lightning activity, possibly an MCS, moved across the area 50-100 km north of Wiener Neustadt in the evening prior to the tornado, 9 July 1916, affecting e.g. Vienna (48 mm of rain and flash floods were reported) and Znoimo in Czechoslovakia. It appears likely that residual cloudiness and/or a particularly low Bowen ration (i.e., a high latent heat and a low sensible heat flux) towards the north intensified the thermal boundary, and the attendant convergence zone, in the morning of 10 July. The first

thunderstorm(s) in the earlier afternoon, prior to the tornadic supercell, could then have completed the work to concentrate the boundary in an ever sharper form close to Wiener Neustadt.

Hailstones in the size of dove eggs or chicken eggs were multiply reported from the vicinity of the tornado. Dörr (1917) mentioned "only" walnut sized hail. It is unclear if he was unaware of these reports, or if he just focused too much on the tornado to pay much attention to the hailfall. There are other reports of hailstones in the size of fists or duck eggs from a handful of places about 20 km upstream of the first tornado evidence, but allegedly at 6.20 pm, i.e. after the tornadic storm. Either the reported time is wrong, or indeed another (supercell) storm followed tracked along the still existing boundary in the wake of the tornadic storm. Either way, there is much more evidence from contemporary reports than the authors describe how conducive this situation was for organized storms and how favourably the ingredients merged for the most extreme development.

In general, I miss a "scale cascade" in the manuscript. Why should we not follow the "forecast funnel" from large to small scales, advocated by Dave Schultz, also in the reconstruction of a historic event?

(3) The paragraphs of the introduction appear like they were shaken by a random generator. As a result, the reader is left rather clueless about the contemporary state of knowledge of the 1916 tornado and about the authors' motivation to reopen this case. The listed "aims" (p. 3, lines 9-13) are no aims but rather research questions. Luckily the rest of the manuscript adresses many more aspects than just these questions, otherwise it would be very meagre. Nonetheless I recommend a reformulation of the entire introduction, and of the aims / research questions in particular.

(4) I don't think Fig. 1 is necessary. It basically shows that many historic tornadoes are known where the population density is high, and few where the population density is low, but this information is not essential. I think it would suffice to mention that

tornadoes (also strong ones) did occur in this area, and that information on them is stored in ESWD.

(5) The text contains a lot of place names, which sometimes make it difficult to follow for someone not familiar with this area. I would suggest to remove names not ultimately necessary for an understanding, and to insert a map which provides an overview of all referred locations at one glance at the beginning of Section 3. Ideally, it would also contain a height shading in order to emphasize the geographic and topographic characteristics of the target area, which would make some of the conclusions more comprehensible.

(6) The map in Fig 3. is very informative but much too specific for such an early placement in this manuscript. Again following a "scale cascade", it should rather find its place somewhere behind Fig. 6. What is the difference between buildings shaded in white ("damage"), dark grey ("minor damage") and black ("major damage") in Fig. 3? The legend does not make this clear. Please remove the TORRO scale references in the legend, as you do not address and explain the TORRO scale anywhere in the text body.

(7) I would not introduce two "precedessor tornadoes", which might mislead a superficial reader to assume that three separate tornadoes occurred on that afternoon. I don't see enough evidence that the two "precedessor tornadoes" could have dissipated, or done anything more mystic and less prosaic, before the main tornado emerged. I would rather call the entire west-east moving vortex the "main tornado", regardless whether its track might have been interrupted somewhere in-between, and the north-south moving subvortex a "satellite tornado" that was finally ingested by the main tornado.

(8) I see the DI-DoD approach as a very useful approach and an important achievement on the path towards an International Fujita scale. This could be emphasized more strongly! The single sentence at the end of the manuscript might easily get lost to the reader's mind, even more as it does not appear connected to the rest of the text.

[Figure]

I think this study is very interesting, and I hope the authors attend to my remarks. I am looking forward to seeing the final version achieved!

Georg Pistotnik (ZAMG)

---

## Author Comment (AC1) · 13 Dec 2017

We are very thankful to the work of the two reviewers. We would also like to thank Georg Pistotnik for his elaborate online comments. The reviews and comments led to fruitful discussions within the team of authors and will help to improve the paper, as we hope. Before we are going to answer the single raised points, we would like to summarize our understanding of the synopsis from reviews and comments: A) We see that a multi-disciplinary paper faces the challenge to be seen as such. Expectations that arise from a paper title and abstract in different disciplines need to be addressed. In our case the paper touches in different aspects at least meteorology, engineering, climatology, geography, social sciences and historical sciences. B) The term "reanalysis" used in the paper title in meteorology is mainly used for numerical reanalyses,

while it has a much broader meaning in other disciplines. Our aim was to use this title in a broad, not meteorology-centric way, which was not understood by those readers, who seem to be meteorology-focused. We therefore consider resubmitting the paper with a slightly altered title: "Forensic Re-Analysis of one of the deadliest Tornadoes in European History and its implications". Only the word "forensic" would be added, pointing towards the sub-discipline of forensic meteorology, where meteorological past events would be reconstructed with strong ties into other sciences. C) The larger-scale weather situation, the environment of the storm that spawned the tornado, was not in the focus of our work. We therefore would like to cut this aspect out of the paper and maybe report on that part in a later, separate paper. This would allow a more stringent structure and would enable us to highlight the main work that has been done in establishing a repeatable research methodology for damage assessment as well as path and magnitude reconstruction of historical tornado and local windstorm cases. D) In our detailed answers below we believe that we can either solve the raised issues or in other cases argue that they are not appropriate.

Anonymous Referee #1 Review of "Re-Analysis of one of the deadliest Tornadoes in European History and its implications" I want to like this paper a lot and recommend publication. The analysis of the damage classification was obviously thorough, and the authors are well-intentioned by compiling these various sources into a single database for analysis. Unfortunately, the paper falls well below expectations. I found the text extremely difficult to follow. In my view, the current poor state of the manuscript shows a disregard for the volunteer time of the reviewers who are forced to try to make sense and provide detailed comments to improve a manuscript. Such revisions are the responsibility of the authors, not the reviewers. In other words, it is not up to us to clean up your mess. Sorry for being so frank, but I am not pleased with the state of this manuscript. A paper should make it clear to the reader what its goal is. That is nowhere clear. The abstract vaguely talks about "demonstrating feasibility", but it is not clear of what. This paper needs a clear single-sentence statement: "The purpose of this paper is to. . .". If the authors can't

do this, then there is little justification for this paper. Answer 1: We agree that the abstract should be more concise. Answer 2: Regarding the purpose of the paper we will add a sentence like: "The main aim of the paper is a detailed investigation of a severe windstorm case in Wiener Neustadt in 1916 using original damage information sources and current knowledge. This aim is fulfilled by answering following research questions: a) What kind of original sources can be found? b) Was the catastrophic windstorm event a tornado? c) How strong was the event on nowadays (tornado) damage rating scale? d) Is it ultimately possible to draw a coherent picture of the entire event?" All these 4 research questions are explicitly answered in section 6 of the paper. We see the value of this paper in the multidisciplinary approach. Readers with expectations towards a single-disciplinary approach might be dissatisfied for the same reason. After the text is made more clear, the paper could use a thorough proofreading by a native English speaker to make it grammatically correct. Answer 3: Regarding native English: In personal communication with one of the executive editors it became clear that NHESS does not expect English on the level of a native speaker at first hand. Moreover, the required level of English is checked before acceptance of the manuscript for discussion. This check was passed. NHESS will provide English proofreading support in the final stage before acceptance. From a practical perspective it would be a quite harsh requirement for most Europeans to ask for native English level, if there are no native speakers at hand in a given research group. At the end we are going to look into the level of English to make sure it satisfies the requirements of the journal. Finally, I will add that the paper was made even more difficult to read because the text lacked indented paragraphs or vertical spaces between paragraphs. Recommendation: Reject and resubmit after substantial revision, rewriting, and proofreading. Answer 4: We strictly followed the formatting guidelines of NHESS. In addition see A) to D) above for general statements.

Abstract: Lines 8-10: The first sentence of an abstract should be clear. This is difficult to read and understand. What are the authors trying to say? "Widen the data basis"? What does this mean? "It could be speculated": By whom? Abstracts should be one

paragraph. Rather than saying "After presenting the methodology", the authors should describe the methods and data. "complex thunderstorm activity in the study area": This description is inadequate as a sound meteorological description of the event. "Complex" says nothing meaningful. Untypical should be atypical. "In the outlook": Outlook of what? The reader has no context for what this means. Clearer writing is needed. Rather than say "we stress the side-benefits of the given study", say explicitly what they are. A reader might not bother reading the entire article, so you should leave them with the most important results in the abstract. Answer 5: We agree that the most important results need to be added to the abstract. We also agree that the first sentence needs to be written much clearer.

On this and other topics. Introduction: The introduction is poorly organized. It's a list of thoughts put together with no coherence and no argument that it tries to advance. Line 24: "Before this study, on a regional level, and in scientific literature also on a national level, it was well known that". This content is entirely unnecessary. It is verbose, wastes readers' time, and does not encourage people to read further. Answer 6: Introduction will be reworked, including clearer statement of the main goal of the paper and more information about the previous research on historical tornado cases. First paragraph: Starts out talking about the event of 1916, but then continues to talk about the terminology. A paragraph should have one consistent topic and theme. The reader has no context for the windstorm of 1916. Develop that topic first, then discuss what the terminology means. Page 2: Lines 21-24: Why is this connected to the text before or after? This paragraph just hangs there. So what does the reader need to know about the surrounding data for the tornadoes? Why does this relate to the 1916 event? Answer 7: This part is of interest here, because there was no clear term for tornado in German language at that time, a historical language problem that might per se not be of interest for meteorologists, but it is the reason why the question needs to be answered whether the event was a tornado or not (see research questions in "Answer 2"). Presenting the local tornado climatology in a revised version will be linked to the presentation of the research on tornadoes in Europe (Groenemeijer – Kuhne and

Antonescu et al.), after stating that these authors have used data from ESWD.

Page 3: Paragraph starting at line 6 is just one sentence. This is improper in a scientific journal article. I got frustrated in section 2 with the poor quality of the writing and lack of organization. As such, more detailed comments have not been provided. I am happy to consider providing more thorough comments ONLY AFTER SUBSTANTIAL REVISIONS FOR READABILITY HAVE BEEN PERFORMED. Answer 8: We think that the structure of the methodology section is as clear as can be in a non-trivial working process, and we cannot relieve readers from this without risking that either the methodology is misunderstood or not documented at all. To our best knowledge the steps needed for a thorough post-analysis of a historical event have never been documented, and therefore we think that this is appropriate here.

Other: 1. Why is an international Fujita scale needed? What is wrong with the existing system that a simple modification to account for other damages be added? Answer 9: There is no existing international system for non-US-standard building practices. The "existing system" (F- or EF-Scale) is often misused in a way that in case the F- or EF-Scale is used outside the US, US-specific assumptions of the F- or EF-Scale are not addressed. Please also see comments of reviewer 2.

2. What is the DI-DoD approach? Is this unique terminology and an approach used by this study? Or, has it been employed previously? More needs to be said about this. In its present state, it is just introduced and the reader is assumed to know what it means. Answer 10: There is a well-referenced introduction of the DI-DoD approach in section 2.3. The anonymous reviewer did not want to read section 2 of the manuscript, as he stated above.

3. P. 5, line 5: Specific page numbers should be provided with direct quotations. Answer 11: We will provide the exact page number of Fujita's memoir book which we are quoting.

4. There appear to be only two new figures (Figs. 3 and 5), but each have a different

purpose. Is the purpose of the paper to re-examine damage classifications (Fig. 3)? Or, is it to describe the meteorology (Fig. 5)? Answer 12: Fig. 3 will be shifted to the results section of the paper. Fig. 5 will be skipped, please refer to general remark C of this answer (see above).

5. I found the general meteorological description in section 3 lacking. What is its purpose? I don't find that it is necessary to this paper. If I am wrong, then the authors need to make more clear why it is important. Again, this issue relates to the purpose of the paper, which is poorly stated, if at all. Answer 13: Thank you for this comment. Please refer to general remark D of this author's response (see above).

6. Reference is made repeatedly to "thunderstorms", but I don't see any reference to cloud-to-ground lightning reported by observers. Are these really thunderstorms, or just convective storms (no lightning or thunder observed)? Answer 14: See section 3.1 of the manuscript: "In the area northwest and around Wiener Neustadt all weather stations reported thunderstorms during the afternoon and early evening hours, many of them together with gale-force wind gusts and hail." 7. References to "modern theory" citing a single paper from 1993 is inappropriate. In fact, the whole manuscript is lacking the most relevant and recent research on convective storms and tornadogenesis. Specifically, not a single paper by Markowski has been cited. Answer 15: This will not be an issue in a revised manuscript. Please refer to general remark C of this author's response (see above). In general, references to new papers are only useful if the research questions are related. In this historical case by far we do not have data in the necessary resolution to compare with findings by Markowski. We were only able to compare to environmental conditions and statistical studies of proximity soundings, the reason for the chosen references. But: We will expand the Introduction section with including more research that has been done on the historical tornado cases.

8. Section 5 is poorly motivated and lacks a coherent argument for the needs for these tools and approaches. Consider the types of papers written by Carles Doswell that show depth of insight and clarity in thinking about the need for European forecast

improvements. Such clarity should be mimicked in the present paper, if the authors wish to motivate the European audience to act. Answer 16: During the project phase the authors have been in contact with Chuck Doswell several times, and we are very thankful for his advice over the past years.

9. When I get to the end of the paper, I don't see the take-home message. Answer 17: An update to the summary section will be made with regards to the improved version of the abstract and the introduction, where goals of the paper will be mentioned more explicitly.

10. Acknowledgements should be one paragraph, not separate sentences. Answer 18: This would be easy to do.

References: These two references to papers from ESSL have been omitted and should be considered to be cited in the revised manuscript. Groenemeijer, P., T. Púcik, A. M. Holzer, B. Antonescu, K. Riemann-Campe, D. M. ĔĞ Schultz, T. Kühne, B. Feuerstein, H. E. Brooks, C. A Doswell, H.-J. Koppert, and R. Sausen, 2017: Severe convective storms in Europe: Ten years of research and education at the European Severe Storms Laboratory. Bull. Amer. Meteor. Soc., doi: 10.1175/BAMS-D-16-0067.1. Antonescu, B., D. M. Schultz, A. Holzer, and P. Groenemeijer, 2017: The risk of tornadoes in Europe. Bull. Amer. Meteor. Soc., 98, 713–728, doi: 10.1175/BAMS-D-16- 0171.1. Answer: 19: Thank you for this suggestion.

---

## Author Comment (AC2) · 13 Dec 2017

We are very thankful to the work of the two reviewers. We would also like to thank Georg Pistotnik for his elaborate online comments. The reviews and comments led to fruitful discussions within the team of authors and will help to improve the paper, as we hope. Before we are going to answer the single raised points, we would like to summarize our understanding of the synopsis from reviews and comments: A) We see that a multi-disciplinary paper faces the challenge to be seen as such. Expectations that arise from a paper title and abstract in different disciplines need to be addressed. In our case the paper touches in different aspects at least meteorology, engineering, climatology, geography, social sciences and historical sciences. B) The term "reanalysis" used in the paper title in meteorology is mainly used for numerical reanalyses,

while it has a much broader meaning in other disciplines. Our aim was to use this title in a broad, not meteorology-centric way, which was not understood by those readers, who seem to be meteorology-focused. We therefore consider resubmitting the paper with a slightly altered title: "Forensic Re-Analysis of one of the deadliest Tornadoes in European History and its implications". Only the word "forensic" would be added, pointing towards the sub-discipline of forensic meteorology, where meteorological past events would be reconstructed with strong ties into other sciences. C) The larger-scale weather situation, the environment of the storm that spawned the tornado, was not in the focus of our work. We therefore would like to cut this aspect out of the paper and maybe report on that part in a later, separate paper. This would allow a more stringent structure and would enable us to highlight the main work that has been done in establishing a repeatable research methodology for damage assessment as well as path and magnitude reconstruction of historical tornado and local windstorm cases. D) In our detailed answers below we believe that we can either solve the raised issues or in other cases argue that they are not appropriate.

G. Kopp (Referee) gakopp@uwo.ca General Comments This is a well-written paper which develops a coherent methodology for assessing tornado intensity and track details from historical sources, and then applying the method to a historically significant event. All tornado archives use reanalysis of historical events, so clarifying the methodology is of importance for users of such databases, as is data storage for future users. Answer 20: We thank the second reviewer for this positive view.

In addition, from an international perspective, developing a clear description of what tornado damage for the most severe European tornadoes looks like is important, since it will contrast with the distinctly different damage in the United States and elsewhere, where building practices and styles differ (e.g., wood-frame houses are the primary indicator for severe tornadoes in North America where the Fujita Scale originated). Such work will ultimately be important for establishing a unified and consistent international

standard for tornado intensity estimation. Answer 21: We fully agree and will consider mentioning this in the introduction. It is important to review the historical cases, because these violent events are very rare in Europe and difficult to compare with the effects of violent tornadoes in North America.

Scientific Questions There is significant uncertainty with the authors' analysis, which I believe they have addressed with reasonable effectiveness. However, there are a couple of ways in which the analysis could be extended to reduce the uncertainty. The F4 rating of the tornado is centred on a brick building with 1m thick walls. It is not clear if this was typical practice, then or now, but one wonders if the capacity of such a wall system could be estimated from current engineering practice or literature. Such an analysis could ultimately provide support for the current rating (or a different one), when combined with appropriate aerodynamic data. The aerodynamics of a wall system, after the roof has been removed, are straightforward and unlikely to be altered much by details of the tornado vortex structure and wind field. I am not suggesting that the authors have to conduct this analysis; however, this aspect of tornado-intensity estimation is not mentioned in manuscript even though it is useful and becoming common amongst engineering analyses of tornado damage. Answer 22: We thank the second reviewer for this suggestion. We will mention it in the limitations that the F4 classification is based in a small number of objects.

A second technical aspect, which is typically important in severe tornadoes is wind-borne debris. I wonder, particularly, about the effects on trees. In severe tornadoes, one typically sees trees that are shredded by the debris (at least in North America), but Figure 9 does not appear to indicate that. Once wonders if observations are available, but are just not reported by the authors. Answer 23: Figure 9 in its upper part already shows the outer parts of the tornado track with less intense wind speeds. This is also supported by the presence of a nearly untouched roof in the upper right. We will try to address this topic more specifically in a revised version of the manuscript by naming in the results specific damage indicator and degree of damage pairs in the region of

highest interest.

Technical Corrections P.2, line 10. Building aerodynamics and structural ′ analysis are much further advanced nowadays as well. âAËŸ c Answer 24: Thank you, we will take this into account in a revised version of the manuscript.

P.4, lines 20-21. ′ Unclear/awkwardly-worded sentence. âAËŸ c Answer 25: We will rephrase the sentence.

P.4, line 30 and following. One could argue that the DI/DoD approach arose with the EF-Scale, not the original Fujita Scale, although the authors are treating the DIs in a simple way that is perhaps more consistent with the Fujita-Scale than that currently used in the EF-Scale. A sentence or two on this would be helpful. Answer 26: We will mention this in the article.

P.6, lines 1-2. Unclear/awkwardly-worded sentence. Answer 27: We will rephrase the sentence.

P.6, line 3. A sentence or two about the meaning and interpretation of "damage prevalence" would be helpful. Answer 28: We will include this.

P.7, Figure 3. It would be helpful to have the track ′ boundaries identified on this map. Answer 29: Thank you for this good suggestion. A new map with the boundaries of the tornado track will be available in the revised manuscript.

P.18, the sentences around line 10. No need to ′ repeat the text from earlier in the manuscript. Answer 30: We will review this part.
* * *

---

## Author Comment (AC3) · 13 Dec 2017

G. Pistotnik georg.pistotnik@zamg.ac.at This is not a full review but a list of comments which, I hope, are helpful for the authors in completing their study. (1) A comment on additional information not yet included: Contemporary newspaper reports uniformly state that 39 fatalities of the tornado were buried at the cemetery of Wiener Neustadt two days after the tragedy. This number is remarkably consistent in all newspapers, though many of them had sent their own correspondents to the site of the tragedy. It can therefore be assumed that this number also had an official origin (possibly "Korrespondenz Wilhelm", then the official organ of the empire, though this was not mentioned anywhere). There is no obvious explanation for the discrepancy between this piece of information and the "official" police list with

32 fatalities referred to in this paper, but at least the possibility should be taken into account that the police list could have been incomplete. Most notably, the list on the memorial stone depicted in Fig. 10 - if I understand it correctly that it is identical with the police list - contains mostly German and a few Slavic names, reflecting the blend of population near the capital of the Habsburg empire, a melting pot between "indigenous" people and migrants attracted from its Slavic crown lands. However, contemporary sources state that a large number of workers in the locomotive fabric, and presumably other places nearby, were Russian prisoners of war. Given that a considerable number of Russian citizens must have been exposed to the tornado when it reached its peak intensity, it would be highly surprising if none of them had fallen victim to its furor. The IMO most plausible explanation is that the Russian victims of the tornado were also among the 39 people buried during the solemn funeral, but stayed anonymous in this process and, in particular, were not filed by the police. There is likely no way to prove this assumption right or wrong anymore, but I think some considerations like these should be addressed. In addition, contemporary newspaper reports also include at least half a dozen other detailed eye-witness reports, which were not quoted in this study but contain valuable information about the tornado's track, width, damage marks, and other characteristics. Answer 31: Thank you for these additional (but probably mostly secondary) sources that we will try to review before we are going to revise our manuscript. In the sources we found it was explicitly mentioned that news of higher numbers of victims that were spread in Vienna are lacking any basis. In case the material has already been collected by you (e.g. in the form of PDF files), we would be glad if you could offer these files to us in order to save time.

In particular, there are two reports with detailed information from Dreistetten, the village where the tornado track started. They also include an additional victim, a 12-year old boy lifed up and killed by the tornado in Dreistetten, which is not mentioned in this manuscript yet. These yet lacking information sources may seem minor in comparison to the huge amount of de-archiving the authors have already done (which I explicitly appreciate!). However, this study is still incomplete as long as they are not included.

Playing the devil's advocate, one could state that the authors already fail when addressing the first of their research questions (p. 3, line 10): "What kind of original sources can be found?" All of these newspaper reports are readily accessibe at the Austrian National Library. The additional amount of work required to browse them and to process their information is limited. Answer 32: We are quite sure that secondary sources can hardly ever be completely collected for such a case, as even newspaper reports in other languages and countries are possible. Following the rules of media research, the content of additional information typically is very limited, if not misleading, because of increasingly anecdotal nature of such retold pieces of information. Assessment of the trustworthiness of such sources is difficult in most cases. Also "correspondents" of newspapers cannot per se be seen as reliable sources. Regarding the Dreistetten case we invested several days to find out what really happened there: The death of a boy is anecdotal, the very probable truth (see manuscript text: "local personal communication (Karl, 2017; Schramböck, 2017" – the head of the local administration knew the injured soldier in person) is that a young soldier was lifted up and severely injured, but he survived.

(2) A comment on the meteorological reconstruction: While I can follow the authors' main conclusions in section 3, it is a bit disappointing to see them interpret the surface wind and pressure field of a 125 km spaced reanalysis (without me wishing to dispute the big value of ERA-20C). This does not live up to their claim of a "forensic" approach which they state several times in the manuscript. It is obvious that not every detail can be reconstructed, not every hypothesis proven 101 years later; this is nothing to be ashamed of. However, weather reports from eye-witnesses, newspapers, and historic weather maps would allow a much closer reconstruction and a much stronger emphasis of potentially important aspects and processes. I am most intrigued by the sharpness of the thermal boundary, shown in Fig. 4. Information on the station height, not given in the manuscript (e.g., one of the 33âŮęC measurements was taken at 850m), reveals that the thermal gradient amounted to >10K across 100 km in terms of potential temperature. It is extremely rare for a preconvective situation in European

mid-summer to feature such a strong thermal contrast. It is unlikely that a synoptic frontal zone alone could have created and sustained it against the influence of strong diurnal heating, which tends to level such differences. Instead, a true forensic analysis should place more emphasis on the "footprints" of previous deep convection prior to the tornadic storm. Weather and newspaper reports mention that strong thunderstorms with heavy rain and very high lightning activity, possibly an MCS, moved across the area 50-100 km north of Wiener Neustadt in the evening prior to the tornado, 9 July 1916, affecting e.g. Vienna (48 mm of rain and flash floods were reported) and Znoimo in Czechoslovakia. It appears likely that residual cloudiness and/or a particularly low Bowen ration (i.e., a high latent heat and a low sensible heat flux) towards the north intensified the thermal boundary, and the attendant convergence zone, in the morning of 10 July. The first thunderstorm(s) in the earlier afternoon, prior to the tornadic supercell, could then have completed the work to concentrate the boundary in an ever sharper form close to Wiener Neustadt. Hailstones in the size of dove eggs or chicken eggs were multiply reported from the vicinity of the tornado. Dörr (1917) mentioned "only" walnut sized hail. It is unclear if he was unaware of these reports, or if he just focused too much on the tornado to pay much attention to the hailfall. There are other reports of hailstones in the size of fists or duck eggs from a handful of places about 20 km upstream of the first tornado evidence, but allegedly at 6.20 pm, i.e. after the tornadic storm. Either the reported time is wrong, or indeed another (supercell) storm followed tracked along the still existing boundary in the wake of the tornadic storm. Either way, there is much more evidence from contemporary reports than the authors describe how conducive this situation was for organized storms and how favourably the ingredients merged for the most extreme development. In general, I miss a "scale cascade" in the manuscript. Why should we not follow the "forecast funnel" from large to small scales, advocated by Dave Schultz, also in the reconstruction of a historic event? Answer 33: Please refer to general comment C above.

(3) The paragraphs of the introduction appear like they were shaken by a random generator. As a result, the reader is left rather clueless about the contemporary state of

knowledge of the 1916 tornado and about the authors' motivation to reopen this case. The listed "aims" (p. 3, lines 9-13) are no aims but rather research questions. Luckily the rest of the manuscript adresses many more aspects than just these questions, otherwise it would be very meagre. Nonetheless I recommend a reformulation of the entire introduction, and of the aims / research questions in particular. Answer 34: Thank you for this comment. We will review the order of paragraphs in section 1. We will also consider renaming the aims into research questions. We will also add more clearly the state of knowledge of 1916, documented in the Dörr 1917 paper, which was primarily a collection of impacts and weather reports, and that there was no magnitude assessment possible at that time.

(4) I don't think Fig. 1 is necessary. It basically shows that many historic tornadoes are known where the population density is high, and few where the population density is low, but this information is not essential. I think it would suffice to mention that tornadoes (also strong ones) did occur in this area, and that information on them is stored in ESWD. Answer 35: We will reconsider this.

(5) The text contains a lot of place names, which sometimes make it difficult to follow for someone not familiar with this area. I would suggest to remove names not ultimately necessary for an understanding, and to insert a map which provides an overview of all referred locations at one glance at the beginning of Section 3. Ideally, it would also contain a height shading in order to emphasize the geographic and topographic characteristics of the target area, which would make some of the conclusions more comprehensible. Answer 36: We discussed a lot about this. Finally we decided to go for the place names, because it would be difficult for readers to look them up in the original sources. We will consider making them better visible in the map of Figure 6.

(6) The map in Fig 3. is very informative but much too specific for such an early placement in this manuscript. Again following a "scale cascade", it should rather find its place somewhere behind Fig. 6. What is the difference between buildings shaded in white ("damage"), dark grey ("minor damage") and black ("major damage") in Fig.

3? The legend does not make this clear. Please remove the TORRO scale references in the legend, as you do not address and explain the TORRO scale anywhere in the text body. Answer 37: Thank you for this suggestion.

(7) I would not introduce two "precedessor tornadoes", which might mislead a superficial reader to assume that three separate tornadoes occurred on that afternoon. I don't see enough evidence that the two "precedessor tornadoes" could have dissipated, or done anything more mystic and less prosaic, before the main tornado emerged. I would rather call the entire west-east moving vortex the "main tornado", regardless whether its track might have been interrupted somewhere in-between, and the north-south moving subvortex a "satellite tornado" that was finally ingested by the main tornado. Answer 38: We want to stay with evidence and keep such speculations out of the paper. We definitely cannot be sure which of the two "predecessor tornadoes" had a continuous track, if one at all, or if even both.

(8) I see the DI-DoD approach as a very useful approach and an important achievement on the path towards an International Fujita scale. This could be emphasized more strongly! The single sentence at the end of the manuscript might easily get lost to the reader's mind, even more as it does not appear connected to the rest of the text. I think this study is very interesting, and I hope the authors attend to my remarks. I am looking forward to seeing the final version achieved! Georg Pistotnik (ZAMG) Answer 39: Thank you, we agree.

---

## Author Response (AR1)

Author's Response

Except for chapter 2 (Methodology), where most parts remained in place, the paper was largely re-written.

All comments by the referees were taken into account. The paper was improved in structure and clarity. English proofreading was performed.

[revised manuscript text omitted]

---

## Referee Report (RR1)

**Second Review of "A forensic re-analysis of the one of the deadliest tornadoes in European history"**

I thank the authors for improving their manuscript by removing the weather information and focusing the purpose of the manuscript around the forensic re-analysis. (Note that this reviewer did not have any problem distinguishing a forensic re-analysis from reanalysis datasets.) This revision has helped tremendously with the readability of the manuscript.

I am, however, unhappy with their marked-up version of the manuscript explaining simply, "All comments by the referees were taken into account. The paper was improved in structure and clarity. English proofreading was performed." and the manuscript highlighted with broad sweeping red comments. More specifics of the tracked changes from the original submission should be shown for the ease of the reviewers comparing to the original manuscript.

1. The authors have improved the English language somewhat, but it still requires more work. If this is to be done by the copyeditors of NHESS to some standard, then so be it. However, I want to impress upon the authors that before their manuscript is passed to the copyeditors it must pass the reviewers at least once. Reviewers struggling through poor language does not make for happy reviewers. Reviewers must be able to read and understand the manuscript. Failure to be clear and precise, regardless of whether the authors are nonnative English speakers or not, is not an acceptable excuse, in my opinion.

2. That being said, it is clear that more precise proofreading by the authors is necessary. Words and punctuation are inconsistently used throughout the manuscript. For example, sometimes "eyewitness" is one word (correct); sometimes it is spelled as two words (incorrect; p. 7, line 3). Sometimes quotes are normal; sometimes they are inverted (p. 8, line 3; p. 10, lines 6-7; p. 20, lines 4 and 12). Sometimes locations are in quotes (e.g., p. 7, lines 18, 23); other times they are not. Citations are used inconsistently (p. 1, lines 23-25; p. 9, line 7). Errors in an author's last name have been made (e.g., p. 18, line 19). All of these errors are clearly not due to not being a native English speaker. Punctuation is improperly used (e.g., p. 5, line 20). These errors are due to the failure to adequately

proofread the manuscript before submission.

3. Title: The title is vague: "One of the deadliest tornadoes in European history" does not tell the reader anything (and is also verbose). How about adding the date and location of the tornado, such as "**A forensic re-analysis of the one of the deadliest European tornadoes: 10 July 1916, Wiener Neustadt, Austria.**"

4. The authors have an inconsistent use of "tornado" in this manuscript. Sometimes the manuscript refers to one tornado (as in the title). Other times reference is made to two tornadoes that merge to form the Wiener Neustadt tornado (p. 17, lines 23-24). Yet other times the tornadoes are considered a "plausible deduction" (p. 14, line 1). Given the uncertainty involved, more skepticism should be stated each time these two predecessors are discussed, rather than definitive statements. More clarity and consistency throughout the manuscript is required.

5. p. 1, line 27: The term "damage point" is used, but not defined. Please define it.

6. p. 1, line 25: "a study by" is not necessary. Delete it here, as well as other similarly unnecessary phrases throughout the manuscript.

7. p. 1, line 16: Should "was" be deleted?

8. p. 2, line 16: The authors misinterpret the findings of Antonescu et al. (2018). The number of fatalities is not "more than 1000 fatalities". It is more correctly stated with the error bars included: "170–1696 fatalities" because the worst-case scenario could be as small as 170, depending on the assumptions that go into their model.

9. p. 2, line 24 is not a question, so it does not fit into the parallel structure of the rest of the items in that list.

10. p. 2, line 1: Is there any truly objective way to do this? I suggest deleting the word "objective".

11. p. 6, line 4-5: Please rephrase "Keeping this physical nature in mind",

which is unclear.

12. p. 17, line 23-24:  In the summary, the manuscript reads that the tornadoes approached each other at 70°.  As far as I can tell, this is the first that is mentioned of this fact (determined by searching the text for "70"), a fact that is presented visually in Fig. 2, but not specifically mentioned in the text.  New facts should be discussed in the body of the text, not in the summary.

13. More specifically, what kind of weather phenomena would explain two tornadoes meeting at 70° angles and merging into a single violent tornado? Is there any historical precedent in the literature?  More explanation that would help the reader understand the physical and plausible reasoning for this occurrence would be welcome.  Otherwise,

---

## Author Response (AR2)

**Author`s response for minor revision on 7 May 2018:**

We thank both referees for their valuable work.

Please find our answers to the comments for referee # 1 below.

The content-related changes are highlighted and commented also in the attached marked-up version.

**Second Review of "A forensic re-analysis of the one of the deadliest tornadoes in European history"**

I thank the authors for improving their manuscript by removing the weather information and focusing the purpose of the manuscript around the forensic re-analysis. (Note that this reviewer did not have any problem distinguishing a forensic re-analysis from reanalysis datasets.) This revision has helped tremendously with the readability of the manuscript.
I am, however, unhappy with their marked-up version of the manuscript explaining simply, "All comments by the referees were taken into account. The paper was improved in structure and clarity. English proofreading was performed." and the manuscript highlighted with broad sweeping red comments. More specifics of the tracked changes from the original submission should be shown for the ease of the reviewers comparing to the original manuscript.
1. The authors have improved the English language somewhat, but it still requires more work. If this is to be done by the copyeditors of NHESS to some standard, then so be it. However, I want to impress upon the authors that before their manuscript is passed to the copyeditors it must pass the reviewers at least once. Reviewers struggling through poor language does not make for happy reviewers. Reviewers must be able to read and understand the manuscript. Failure to be clear and precise, regardless of whether the authors are nonnative English speakers or not, is not an acceptable excuse, in my opinion.

2. That being said, it is clear that more precise proofreading by the authors is necessary. Words and punctuation are inconsistently used throughout the manuscript. For example, sometimes "eyewitness" is one word (correct); sometimes it is spelled as two words (incorrect; p. 7, line 3).

Sometimes quotes are normal; sometimes they are inverted (p. 8, line 3; p. 10, lines 6-7; p. 20, lines 4 and 12).

All these errors were corrected.

Sometimes locations are in quotes
(e.g., p. 7, lines 18, 23); other times they are not.

If place names are only available in local dialects and/or not represented in official maps, locations are in quotes.

Citations are used
inconsistently (p. 1, lines 23-25; p. 9, line 7).

It depends if directly mentioned in the main text or not.

Errors in an author's last name
have been made (e.g., p. 18, line 19).

Corrected.

All of these errors are clearly not
due to not being a native English speaker. Punctuation is improperly used
(e.g., p. 5, line 20).

Corrected.

These errors are due to the failure to adequatelyproofread the manuscript before submission.

3. Title: The title is vague: "One of the deadliest tornadoes in European history" does not tell the reader anything (and is also verbose). How about adding the date and location of the tornado, such as "**A forensic reanalysis of the one of the deadliest European tornadoes: 10 July 1916, Wiener Neustadt, Austria.**"

Part of the suggestion was taken over. The long version would be verbose.

4. The authors have an inconsistent use of "tornado" in this manuscript. Sometimes the manuscript refers to one tornado (as in the title). Other times reference is made to two tornadoes that merge to form the Wiener Neustadt tornado (p. 17, lines 23-24). Yet other times the tornadoes are considered a "plausible deduction" (p. 14, line 1). Given the uncertainty involved, more skepticism should be stated each time these two predecessors are discussed, rather than definitive statements. More clarity and consistency throughout the manuscript is required.

We are using the words "likely", "plausible" and "possible" to describe the uncertainty with the discussed scenarios.

5. p. 1, line 27: The term "damage point" is used, but not defined. Please define it.

We made the reference to the Johns et al paper clearer.

6. p. 1, line 25: "a study by" is not necessary. Delete it here, as well as other similarly unnecessary phrases throughout the manuscript.
7. p. 1, line 16: Should "was" be deleted?
8. p. 2, line 16: The authors misinterpret the findings of Antonescu et al. (2018). The number of fatalities is not "more than 1000 fatalities". It is more correctly stated with the error bars included: "170–1696 fatalities" because the worst-case scenario could be as small as 170, depending on the assumptions that go into their model.
9. p. 2, line 24 is not a question, so it does not fit into the parallel structure of the rest of the items in that list.
10. p. 2, line 1: Is there any truly objective way to do this? I suggest deleting the word "objective".
11. p. 6, line 4-5: Please rephrase "Keeping this physical nature in mind", which is unclear.

All corrected.

12. p. 17, line 23-24: In the summary, the manuscript reads that the tornadoes approached each other at 70°. As far as I can tell, this is the first that is mentioned of this fact (determined by searching the text for "70"), a fact that is presented visually in Fig. 2, but not specifically mentioned in the text. New facts should be discussed in the body of the text, not in the summary.

Added on page 9.

13. More specifically, what kind of weather phenomena would explain two tornadoes meeting at 70° angles and merging into a single violent tornado? Is there any historical precedent in the literature? More explanation that would help the reader understand the physical and plausible reasoning for this occurrence would be welcome. Otherwise,

We describe this in section 4.2 on page 9 and refer to the Fujita and Merriam book:

[revised manuscript text omitted]